# Two Decades Later: Long-Term Multisystem Sequelae and Subclinical Organ Dysfunction in Sudan Ebola Virus (SUDV) Survivors of the 2000 Outbreak

**DOI:** 10.3390/v17111410

**Published:** 2025-10-23

**Authors:** Raymond Ernest Kaweesa, Joseph Ssebwana Katende, Geoffrey Odoch, Annie Daphine Ntabadde, Raymond Reuel Wayesu, Deborah Mukisa, Peter Ejou, Pontiano Kaleebu, Jennifer Serwanga

**Affiliations:** 1Department of Immunology, Uganda Virus Research Institute, Entebbe, Uganda; raykaw35@gmail.com (R.E.K.); joseph.ssebwanakatende@mrcuganda.org (J.S.K.); dntabadde@uvri.go.ug (A.D.N.); rsseguya@uvri.go.ug (R.R.W.); deborah.mukisa@mrcuganda.org (D.M.); pontiano.kaleebu@mrcuganda.org (P.K.); 2Medical Research Council/Uganda Virus Research Institute & London School of Hygiene & Tropical Medicine (MRC/UVRI & LSHTM) Research Unit, Entebbe, Uganda; geoffrey.odoch@mrcuganda.org (G.O.); peter.ejou@mrcuganda.org (P.E.)

**Keywords:** sudan ebolavirus, long-term sequelae, Post EVD syndrome, ebola survivors, subclinical dysfunction, psychosocial resilience, viral hemorrhagic fever

## Abstract

**Background**: Despite repeated re-emergence of Sudan ebolavirus (SUDV), its long-term human toll remains under-characterised. We assessed multisystem clinical, biochemical, and psychosocial outcomes ~25 years after the 2000 Gulu outbreak. **Methods**: We conducted a cross-sectional evaluation of 45 survivors of laboratory-confirmed SUDV and 30 age- and gender-matched community controls from the same region. Symptoms were assessed as current at the study visit using a structured checklist; for each symptom present, we recorded severity and duration from onset to the visit date. Standardised clinical examinations, haematological and biochemical assessments, anxiety and depression screening, and structured interviews on social support and stigma were performed. Group comparisons were assessed with Wilcoxon rank-sum and χ^2^/Fisher’s exact tests; correlations were assessed with Spearman’s ρ. **Findings**: Core physiological indices (vital signs, BMI, blood pressure, and body temperature) and mental health were comparable between survivors and controls. Nevertheless, survivors reported ongoing symptoms, including joint pain and visual impairment each in 36% (16/45), fatigue in 18% (8/45), and neurological symptoms in 13% (6/45). Subclinical laboratory deviations centred on hepatic and platelet biology: elevated total bilirubin occurred in 14% of survivors versus 6.7% of controls; thrombocytopenia or platelet morphological abnormalities in 12% versus 3.3%; haemoglobin abnormalities in 6% versus 0%. Among survivors, albumin and mean platelet volume declined with age (both *p* ≤ 0.03). Psychological morbidity was low (normal anxiety 82% (37/45; and normal depression 80% (36/45). Yet a social paradox emerged, despite universal post-outbreak support, 98% (44/45) described enduring stigma. To minimise differential recall bias, symptom inventories were not collected from controls; consequently, between-group comparisons for symptom prevalence were not performed, and symptom inferences are restricted to survivors and framed descriptively. **Interpretation**: A quarter-century after infection, SUDV survivors show preserved systemic physiology but carry chronic musculoskeletal, sensory, and neurological sequelae, alongside a discrete subclinical profile implicating hepatic function and platelet biology. Psychological resilience coexists with near-universal, persistent stigma, indicating that material support did not achieve full psychosocial reintegration. Given the lack of virological and deep immune profiling, proposed pathogenetic mechanisms, such as antigen persistence or immune-mediated injury, remain speculative and hypotheses-generating only. These findings argue for survivor-centred long-term care, embedded with epidemic preparedness frameworks that integrate musculoskeletal rehabilitation, ophthalmic and neurological services with comprehensive mental health care, and sustained anti-stigma community engagement. This dissociation, including short-lived support alongside enduring stigma, indicates that humanitarian relief alone does not secure durable psychosocial reintegration and should be complemented by long-horizon, survivor-centred services and community engagement. **Funding**: This study was supported by the Coalition for Epidemic Preparedness Innovations (CEPI) under the Universal Protocol for Standardising Assays and Advancing Vaccine Immunogenicity Assessments for Emerging and Re-emerging Viral Threats, implemented through the Uganda Virus Research Institute (UVRI) as part of CEPI’s Centralised Laboratory Network (CLN).

## 1. Introduction

Ebola Virus Disease (EVD) remains among the lethal infectious threats in sub-Saharan Africa, characterised by recurrent outbreaks, high case fatality rates, and profound societal disruption. Among the six known *Ebola virus* species, *Sudan ebolavirus* (SUDV) stands out for its repeated epidemics since its first identification in 1976 [1], with case fatality rates ranging from 41% to 100%. Clinically, SUDV infection is indistinguishable from other EVDs, presenting after an incubation period of 2–21 days with non-specific symptoms such as fever, headache, fatigue, myalgia, and pharyngitis, features that overlap with endemic febrile illnesses such as malaria and typhoid [2]. Progression to severe disease may involve gastrointestinal disturbance, mucosal bleeding, petechiae, and neuroinflammatory manifestations including delirium and encephalopathy, all associated with poor prognosis [3,4,5,6,7].

While post-Ebola virus disease syndrome (PEVDS) has been well described among survivors of *Zaire Ebolavirus* (EBOV), manifesting as chronic fatigue, arthralgia, neurocognitive impairment, and ocular disease [8,9,10,11], the long-term sequalae remain of SUDV remain poorly defined. Emerging evidence suggests that filovirus infection induces durable physiological and immunological perturbations, including chronic musculoskeletal pain, sensory deficits, and haematological abnormalities, which persist for years after acute illness [12,13]. Data from the 2022–2023 SUDV outbreak in Uganda highlight multi-systemic post-infection morbidity, including persistent musculoskeletal, neurological, ophthalmic, and respiratory symptoms, with viral RNA detected in semen and breast milk up to seven months post-infection [14]. These findings suggest potential viral persistence and latency, underscoring the importance of structured follow-up for survivors. Importantly, long-term SUDV survivors have shown sustained virus-specific humoral and cellular immunity, with evidence of innate and adaptive activation persisting for more than 15 years after recovery [15].

The recurrent resurgence of SUDV in Uganda, documented in 2000, 2007, 2011, 2022, and most recently 2025 [16,17,18], underscores both its epidemic potential and the lack of investment in survivor science compared with EBOV. The 2000 Gulu outbreak, the largest SUDV epidemic to date, generated a unique cohort of survivors whose long-term health outcomes remain incompletely characterised.

Here, we address this critical knowledge gap by conducting a cross-sectional evaluation of survivors from the 2000 Gulu outbreak. Through detailed clinical, biochemical, and immunological assessments compared with age- and sex-matched community controls, we delineate multisystem sequelae and subclinical organ dysfunction attributable to SUDV. By defining the chronic clinical manifestations and subtle immune-related changes detectable only by laboratory testing of SUDV infection, this study provides essential insights for survivor care, epidemic preparedness, and the development of therapeutic interventions for neglected filoviral diseases.

## 2. Materials and Methods

### 2.1. Study Design and Population

We conducted a cross-sectional study involving two cohorts drawn from Ugandan communities affected by the 2000–2001 SUDV outbreak. The survivor cohort comprised 45 individuals with Ministry of Health documented recovery from laboratory-confirmed SUDV infection during that epidemic. The comparator group consisted of 30 community controls who were age- and gender-matched, recruited from the same geographical region to account for local and socioeconomic confounding factors. These controls reported no history of prior SUDV infection or clinical illness consistent with Ebola virus disease.

All participants provided written informed consent prior to enrollment. The study protocol was reviewed and approved by the Uganda Virus Research Institute Research Ethics Committee (UVRI-REC GC/127/1045) and the Uganda National Council for Science and Technology (UNCST HS5212ES). All procedures adhered to the principals outlined in the Declaration of Helsinki and Good Clinical Practice (GCP) guidelines. Samples were collected, de-identified and stored in secure biorepositories and standardised protocols to enable future immunovirological investigations. To contextualise sample representativeness, we benchmarked our enrolled survivors against published outbreak-level demographics from the 2000–2001 SUDV epidemic (425 cases; 224 deaths; ~201 survivors; case fatality ~53%; ~93% of cases from Gulu district; median age 27 years; 63% female; 31 infected healthcare workers) [19,20]. Our study cohort (*n* = 45) therefore represents ~22% of the estimated survivor pool, with a gender distribution (64% female) and age-at-infection profile (predominantly 20–39 years) aligned with the outbreak-wide figures. We recruited in the same outbreak geography (Gulu and environs), further minimising geographic selection effects.

### 2.2. Data Collection and Clinical Assessment

Sociodemographic characteristics (age category, sex, occupation) and detailed clinical assessments were systematically collected using standardised tools. At the study visit, participants reported current post-Ebola symptoms using a prespecified checklist (fatigue, myalgia, arthralgia, neurocognitive complaints, sensory impairments, gastrointestinal symptoms). For each symptom present, severity (none, mild, moderate, severe) and duration (in years since onset) were recorded. Occupational categories were harmonised post hoc according to international classification systems. Vital signs were benchmarked against reference ranges (heart rate: 60–100 bpm; respiratory rate: 12–20 breaths/min; temperature: 36.1–37.2 °C; BMI: 18.5–24.9 kg/m^2^; blood pressure: systolic 90–120 mmHg, diastolic 60–80 mmHg) [21,22]. Survivors self-reported hallmark post Ebola sequelae, including fatigue, myalgia, arthralgia, neurocognitive dysfunction, sensory impairments, and gastrointestinal symptoms. For each symptom, severity was graded (none, mild, moderate, severe) and duration recorded in years. To reduce differential recall and salience bias, the detailed symptom inventory was not administered to community controls, as they lacked an Ebola-related illness anchor. This decision was made a priori to avoid over- or under-reporting due to differing motivational and contextual frames between survivors and controls. As a result, estimates of symptom prevalence and severity are presented for survivors only, without formal between-group symptom comparisons.

### 2.3. Laboratory and Immunological Evaluation

Venous blood and midstream urine were collected aseptically. Complete blood counts were obtained to characterize hematologic profiles. Renal function was assessed by creatinine and urea, and hepatic function by bilirubin, alanine aminotransferase (ALT), and aspartate aminotransferase (AST). Inflammatory activity was evaluated by quantifying C-reactive protein (CRP). Urinalysis was performed using a standard multi-parameter urine dipstick (Siemens Multistix 10SG, Siemens Healthineers, Erlangen, Germany). Results were recorded per the manufacturer’s colour chart as Negative, Trace, 1+, 2+, or 3+; these are semi-quantitative (ordinal) categories and not absolute concentrations, which vary by strip brand. Laboratory abnormalities were defined using internationally accepted clinical thresholds [21,22]. Peripheral blood mononuclear cells (PBMCs) were cryopreserved to permit future immune profiling.

### 2.4. Mental Health and Social Outcomes

Psychological status was assessed using the Hospital Anxiety and Depression Scale (HADS), a validated screening tool previously applied in cohorts of Ebola and Marburg virus disease survivors [23], which indexes current. HADS was administered at the visit and reflects current anxiety and depression burden at the time of assessment. The instrument categorises symptoms into four severity levels: normal (0–7), mild (8–10), moderate (11–15), and severe (≥16) [24,25]. HADS was administered to both survivors and controls. To capture the broader social dimensions of post-recovery experience, binary indicators documented the presence or absence of perceived stigma (from community or workplace) and support systems (from family, peers, or community networks). In addition to HADS, we captured social experiences using binary indicators for (a) perceived stigma (community or workplace) and (b) perceived support (from family, peers, community networks, faith groups, or material relief). We distinguished social and material support from clinical services: the former included food, cash or in-kind assistance, communal caregiving, and social inclusion; the latter referred to health-facility-based care after discharge. Clinical services were recorded qualitatively via free-text prompts (presence/examples) but were not quantified. We also asked participants whether support was perceived as (i) immediate and time-limited to the outbreak/recovery period or (ii) ongoing at the time of interview. To minimise recall-frame bias, we did not ask for precise start/stop dates or intensity; consequently, duration could not be estimated. This combined approach allowed evaluation of both mental health burden and the social determinants shaping long-term survivor outcomes. The full survivor questionnaire (demographics, structured symptom checklist, and stigma/support items) is provided in Appendix A. For the HADS, we include administration and scoring details with citation to the validated instrument; verbatim items are not reproduced due to licensing.

### 2.5. Statistical Analysis

We compared physiological, biochemical, and psychological outcomes between SUDV survivors and age- and sex-matched community controls. Descriptive statistics were used to summarise demographic and clinical characteristics. Continuous variables were analysed with the Wilcoxon rank-sum test, and categorical variables with Pearson’s χ^2^ test or Fisher’s exact test, as appropriate. Gender-stratified analyses were conducted to assess sex-based differences in physiological parameters, with non-significant comparisons denoted as “ns.” Associations between clinical symptoms and laboratory measures were evaluated using Spearman’s rank correlation. All statistical tests were two-sided, and *p*-values < 0.05 were considered significant. Analyses were performed in R (version 4.2.0, R Foundation for Statistical Computing, Vienna, Austria). No hypothesis tests compared symptom prevalence between survivors and controls because symptom inventories were not collected in controls to avoid recall bias. All symptom analyses are descriptive within the survivor cohort.

In addition to non-parametric group comparisons and adjusted regression models, we computed age-adjusted partial correlations (Spearman) for all vital sign–laboratory pairs within survivors and controls using complete-case analysis. We compared the original and age-adjusted coefficients to quantify the impact of age; summary metrics included the mean absolute change in r (|Δr|) and retention of statistical significance.

## 3. Results

### 3.1. Demographic and Occupational Characteristics of SUDV Survivors Versus Matched Controls

Among 45 survivors, 64.4% were female (29/45; Figure 1A), consistent with caregiving roles that likely increase exposure during the rural outbreak [26]. Most survivors were infected at ages 20–39 years and were 45–64 years at enrolment (Figure 1B). Controls (*n* = 30) were age- and sex-matched at the group level. Individuals aged ≥65 years were uncommon in both groups (six survivors, two controls), potentially reflecting higher outbreak mortality or lower exposure among older adults. Gender–age interaction showed a pronounced female predominance at older ages: male-to-female survivor ratios were 0.54 (25–44 years), 0.71 (45–64 years), and 0.20 (≥65 years). This pattern is directionally consistent with national life-expectancy differences in Uganda (women 70.1 vs. men 66.9 years) and with literature on sex-differentiated antiviral immunity [27,28,29,30].

Occupational histories reflected the rural context: agriculture predominated in both survivors and controls (Figure 1C). However, healthcare workers appeared only among survivors, aligning with early nosocomial exposure before widespread recognition and containment; educators were more common among controls, plausibly due to occupational shielding during school closures. In this 25-year follow-up of the 2000 SUDV outbreak, demographic and occupational profiles point to exposure-driven risk rather than survival bias.

### 3.2. Physiological Restoration 25 Years After Sudan Ebolavirus Infection

Twenty-five years after the 2000 SUDV outbreak, survivors exhibited apparent clinically normal physiology, indistinguishable from that of age- and sex-matched unexposed controls (Figure 2). Core vital and anthropometric measures lay within accepted clinical ranges, supporting durable restoration of autonomic and metabolic function within accepted clinical reference ranges, indicating durable recovery of autonomic and metabolic regulation. Resting heart rate showed wide dispersion, yet no between-group differences (survivors: median 75 bpm [IQR 50–144]; *p* > 0.05 versus controls). Respiratory rate (16 breaths per minute [IQR 14–18]) and body temperature (36.2 °C [IQR 35.2–36.5]) were likewise comparable to controls (all *p* > 0.05) and consistent across gender. Anthropometry indicated comparable nutritional status, with a median BMI of 22.3 kg/m^2^ (IQR 18.1–34.7) in survivors and 22.2 kg/m^2^ (IQR 17.5–37.5) in controls. Elevated BMI (greater than 25 kg/m^2^) occurred in both cohorts, more often among women, suggesting shared environmental or nutritional determinants rather than post-viral effects. Blood pressure profiles were similar between groups and sexes (all *p* > 0.05), with medians trending above conventional thresholds in both cohorts (systolic: survivors 127.0 mmHg [IQR 94.0–164.0], controls 131.5 mmHg [101.0–170.0]; diastolic: survivors 80.0 mmHg [61.0–102.0], controls 81.0 mmHg [60.0–106.0]).

Taken together, these data show preserved physiological integrity and no evidence of chronic autonomic or metabolic dysfunction attributable to prior SUDV infection, indicating survivors can achieve full systemic recovery following convalescence.

### 3.3. Persistent Post-Ebola Sequelae in Sudan Virus Disease Survivors Highlight a Chronic Musculoskeletal, Sensory, and Neurological Burden

Because a structured symptom inventory was not administered to controls to mitigate recall bias, the analyses in this section are limited to survivors and do not include between-group prevalence estimates. Among 45 Sudan virus disease (SVD) survivors from the 2000 Gulu outbreak, self-report and targeted clinical evaluation revealed a notable burden of chronic sequelae (Figure 3A). Joint pain and visual impairment were the most frequent, each reported by 36% (16/45; Figure 3B). Fatigue was reported by 18% (8/45) and neurological symptoms, including paresthesia, peripheral tingling, and cognitive slowing, were reported by 13% (6/45). Severity gradients differed by domain: visual impairment showed the broadest range and included several cases rated as severe and functionally disabling, while joint pain contributed substantially to moderate disability with direct consequences for mobility, productivity, and quality of life. Duration analysis (Figure 3C) indicated protracted courses, with joint and visual complaints commonly persisting for ≥ 5–7 years, and individual cases extending beyond 15 years. High inter-individual variability (SD approaching 10 years) suggests heterogeneous trajectories of recovery and possible focal, chronic inflammation within affected tissues. The anatomical clustering of symptoms in joints, ocular tissues, and the nervous system, including sites with immune-privileged characteristics, aligns with emerging models of post-EVD pathology, which involve persistent antigenic stimulation and immune-mediated tissue injury. While our study did not assay viral material, these patterns are consistent with previous reports in Ebola virus survivors, which describe the presence of residual viral RNA and chronic immune activation in immune-privileged sites [10,11,31]. We did not conduct virological testing or immune phenotyping; therefore, any reference to antigen persistence or immune-mediated injury here is hypothesis-generating rather than evidentiary.

Taken together, these results delineate the long-term clinical legacy of SVD and highlight an urgent need for survivor-centred rehabilitation and long-horizon follow-up, particularly for musculoskeletal, visual, and neurological complications. From a health-systems perspective, decades-long disability underscores gaps in epidemic recovery frameworks and argues for dedicated, sustained services for survivors of neglected tropical diseases.

### 3.4. Subclinical Laboratory Profile Concentrates in Hepatobiliary and Platelet Indices with Preserved Renal Function and Minimal Age Confounding in Long-Term SUDV Survivors

For clarity, we grouped laboratory measures by organ system. Hepatobiliary indices comprise total bilirubin (TBIL), direct bilirubin (DBIL), aminotransferases (ALT, AST), and albumin. Platelet indices include platelet count, platelet morphology, and mean platelet volume (MPV). Renal indices include creatinine, blood urea nitrogen (BUN), and urinalysis dipstick (protein, glucose, ketones, bilirubin, blood, nitrite, leukocyte esterase). Across these domains, most values in both survivors and controls fell within clinical reference limits; differences concentrated in hepatobiliary and platelet biology, while renal function was preserved (Figure 4). Across laboratory domains, values were largely within reference limits in both groups. In the panel of normal–abnormal proportions (Figure 4A), between-group comparisons on the underlying continuous measurements (Mann–Whitney U) identified five analytes with separation: RDW-CV, albumin, HbA1c, urine specific gravity (*p* < 0.05 for each), and haemoglobin within-run coefficient of variation (*p* < 0.001). Survivors displayed a higher proportion of abnormalities for RDW-CV and HbA1c, whereas albumin abnormalities were more frequent among controls; specific gravity differed modestly. The within-run CV of haemoglobin is reported for completeness as an assay-precision metric, rather than a physiological endpoint.

Correlation structure was modest and physiologically conventional (Figure 4B): effect sizes spanned roughly −0.30 to 0.50, with a positive, significant association between age and systolic blood pressure and a negative, significant association between diastolic blood pressure and serum chloride; the remaining prespecified pairs were non-significant and did not suggest divergent coupling between survivors and controls. Adjustment for age produced minimal change in these associations (Figure 4C), with points clustering tightly along the 45° identity line, indicating negligible attenuation or strengthening of the raw coefficients after age control and arguing against age as the primary driver of the limited between-group signals.

Dipstick urinalysis profiles were reassuring in both cohorts (Figure 4D). Nitrites were almost uniformly negative; protein, glucose, ketones, and bilirubin positivity were rare to absent. Low-grade leukocyturia (trace/1+) occurred in a minority at a similar frequency across groups, and haematuria (blood) was uncommon, appearing slightly more frequent among survivors on visual inspection. Taken together (Figure 4A–D), the data indicate a broadly normal laboratory status 25 years after infection, with a narrow set of test-specific differences and correlation patterns that remain stable after adjusting for age.

### 3.5. Psychological Resilience and the Paradox of Social Support Among Ebola Sudan Virus Survivors

Against expectations of lasting psychological morbidity after Ebola, HADS screening at the study visit showed largely reassuring profiles. Among survivors, anxiety was normal in 82% (37/45), mild in 13% (6/45), and severe in 4% (2/45), Figure 5A; depression was normal in 80% (36/45), mild in 18% (8/45), and severe in 2% (1/45), Figure 5B. In controls, all participants (30/30) scored within the normal range for both anxiety and depression. Thus, any non-normal HADS scores occurred exclusively among survivors, although absolute levels were low and the distributions remained dominated by ‘normal’ in both groups. However, social reintegration findings revealed a paradox (Figure 5C): all survivors (45/45) reported comprehensive community support immediately after the outbreak, yet 98% (44/45) described persistent stigma, commonly linked to fears of viral persistence and transmission. Visible acts of care, therefore, did not equate to psychosocial acceptance; psychological recovery proceeded alongside enduring social exclusion, a gap that current post-outbreak models fail to address.

Survivors uniformly reported receiving post-outbreak support, which they most commonly described as immediate and time-limited social or material assistance (e.g., family/community caregiving, in-kind aid, faith and/or community solidarity). By contrast, structured long-term clinical services after discharge were inconsistently described in free-text responses and were not captured quantitatively in this study. Despite these visible acts of care, stigma persisted for nearly all survivors, often framed around fears of viral persistence and contagion. Because we did not collect precise timing or intensity of support, we cannot quantify duration; however, the prevailing narrative was that support waned over time whereas stigma remained enduring.

## 4. Discussion

A quarter of a century after the 2000 Gulu epidemic, this cohort of Sudan virus disease (SUDV) survivors shows that apparent physiological normalisation can coexist with a durable burden of clinical sequelae and a paradoxical social afterlife of disease. Most physiological and biochemical measures lay within clinical normal reference ranges in both groups, while group differences concentrated in hepatobiliary and platelet indices suggest a subclinical phenotype compatible with long-term perturbations in these systems. We document sustained musculoskeletal, visual, and neurological morbidity alongside subclinical haematological and hepatic alterations, contrasted with broadly preserved vital signs and metabolic indices and unexpectedly robust mental health profiles. Together, these findings show that recovery from SUDV is neither binary nor time-limited: survivors can regain physiological stability yet continue to accrue organ-specific and social costs long after the outbreak has faded from view.

The juxtaposition of short-lived post-outbreak support with long-term stigma underscores a mismatch between relief-oriented responses and the survivor’s lived social trajectory. In our data, support was predominantly immediate and time-limited, centred on social or material relief rather than sustained clinical follow-up, while stigma, anchored in fears of infectivity, remained near-universal years later. This support–stigma dissociation helps explain why acceptable HADS scores can coexist with reported social exclusion: clinical stability is necessary but insufficient for psychosocial reintegration.

The symptom constellation we observe, joint pain, visual disturbance, and neurological complaints, converges with reports among Zaire ebolavirus (EBOV) survivors, extending the evidence base from five-year horizons to 25 years and from EBOV to SUDV [10,11]. The anatomic distribution of morbidity, often localised to immune-privileged or difficult-to-access compartments (joints, ocular tissues, nervous system), is consistent with but does not prove, models invoking antigen persistence and immune-mediated tissue injury [7,32], and aligns with prior detection of viral RNA and chronic immune activation in such sites [10,11,33], our study did not include virological assays or deep immunophenotyping and cannot adjudicate mechanism. We therefore frame these pathogenetic interpretations as hypotheses that warrant prospective testing. In parallel, we identify a pattern of symptom-free laboratory changes centred on platelet indices and hepatic biochemistry, as well as thrombopoietic/endothelial perturbation and abnormalities in bilirubin or albumin levels, despite otherwise reassuring renal and metabolic profiles. While causal mechanisms cannot be inferred in this cross-sectional design, the pattern is biologically plausible after a systemic viral haemorrhagic fever, and merits targeted mechanistic follow-up.

Notably, psychological screening revealed low levels of anxiety and depression, diverging from some EBOV literature emphasising post-traumatic stress and psychological distress [33]. Several factors could contribute, including resilience processes, social coping strategies, and differences in the epidemic context. However, the near-universal persistence of stigma despite reports of early community support exposes a dissociation between visible acts of care and deep psychosocial acceptance. Fears of infectivity and contagion appear to be enduring social constructs that persist beyond clinical recovery. This stigma–resilience paradox underscores why clinical scores alone under-estimate the survivor burden and why reintegration must be an explicit objective of post-outbreak policy.

Our design offers several strengths, including a uniquely long follow-up interval, group-level, age- and sex-matched community controls, and a multidimensional assessment spanning clinical, laboratory, and psychosocial domains, which helps separate SUDV-related signals from background endemic conditions and ageing. At the same time, limitations deserve emphasis. Our cross-sectional design precludes causal inference and is subject to recall bias for self-reported outcomes; and unmeasured comorbid or environmental exposures may contribute to subclinical laboratory deviations. To limit recall or salience bias, controls did not complete a detailed symptom inventory; accordingly, between-group estimates of symptom prevalence or specificity are not possible, and symptom data should be interpreted as survivor-only descriptors rather than population-level contrasts. We recognise that HADS reflects current symptoms at the time of assessment and does not quantify cumulative distress since the outbreak; moreover, with zero non-normal HADS scores in controls and modest sample size, formal between-group hypothesis testing is underpowered and was not emphasised. Also critically, we did not perform virological testing (e.g., RNA/antigen detection in blood, semen, ocular fluid, or other compartments) or immunophenotyping (e.g., cellular activation or exhaustion profiling, cytokines, or single-cell analyses); consequently, references to antigen persistence or immune-mediated injury are necessarily speculative. Unmeasured comorbid or environmental exposures may contribute to the observed subclinical laboratory deviations. Symptom inventories were not collected from unexposed controls to avoid incomparable recall frames, which focuses inference on survivor biology but precludes direct symptom-prevalence comparisons. Future studies should adopt longitudinal designs with harmonised, neutrally framed symptom capture in both survivors and controls (e.g., anchoring vignettes or prospective diaries) to balance recall while enabling valid between-group comparisons. Also, we lacked historical medical records for longitudinal medical verification and did not systematically obtain objective corroboration of specific symptoms (e.g., formal ophthalmic examination, musculoskeletal imaging); thus, the onset and duration are based on self-report and are subject to recall bias. The cross-sectional design precludes estimation of incidence or resolution trajectories; future work should include longitudinal follow-up with standardised objective measures and record abstraction where available. Finally, although we addressed age by matching and by age-adjusted correlations, we did not conduct formal group × age interaction or age-stratified analyses; consequently, we cannot exclude residual age effects on albumin and MPV. Building on these hypothesis-generating findings, we recommend longitudinal follow-up of survivors incorporating (i) targeted virological testing of ethically accessible compartments implicated in post-SUDV morbidity, including ocular assessments with clinical indication, and semen where appropriate, (ii) ophthalmic examination and imaging, musculoskeletal evaluation, and neurocognitive testing. For this cohort, deep immunophenotyping of our cryopreserved PBMCs is planned, including flow cytometry, cytokine profiling, single-cell transcriptomic, and immune-repertoire analyses to evaluate sustained immune activation, and potential links to clinical outcomes. Sensitivity analyses demonstrated minimal age confounding of key associations (all significant correlations retained after age adjustment; mean |Δ*r*| ≈ 0.02), supporting that the hepatobiliary/platelet profile reflects survivor biology rather than age alone. Although we addressed age by matching and by age-adjusted correlations, we did not conduct formal group × age interaction or age-stratified analyses; consequently, we cannot exclude residual age effects.

The policy implications are immediate. To close the gaps, survivor care requires institutionalisation as a long-term health and social protection, not a short-term humanitarian add-on. In filovirus-endemic settings, survivor clinics should integrate musculoskeletal rehabilitation, ophthalmic and basic neurology services and routine laboratory panels (hepatic and platelet-endothelial biomarkers) with ongoing mental health care. Crucially, health systems should pair care with community level anti-stigma communication that addresses misconceptions about infectious risk, co-design reintegration activities with survivors and local leaders, and ensure scheduled follow-up beyond the emergency phase. Research priorities include longitudinal cohort follow-up with imaging, ophthalmic examination, and multi-omics to test hypotheses of antigen persistence and immune dysregulation; interrogation of endothelial and platelet biology; and evaluation of targeted interventions that could modify chronic inflammation and improve function.

In conclusion, this study extends the evidence base for the long-term consequences of SUDV, demonstrating that survivors can achieve physiological stability while carrying a persistent clinical and social burden. Closing the gap between recovery and restored health will require survivor-centred care models, stigma-reducing community strategies, and mechanistic research that can seed therapeutic innovation for neglected filoviral diseases.

## Figures and Tables

**Figure 1 viruses-17-01410-f001:**
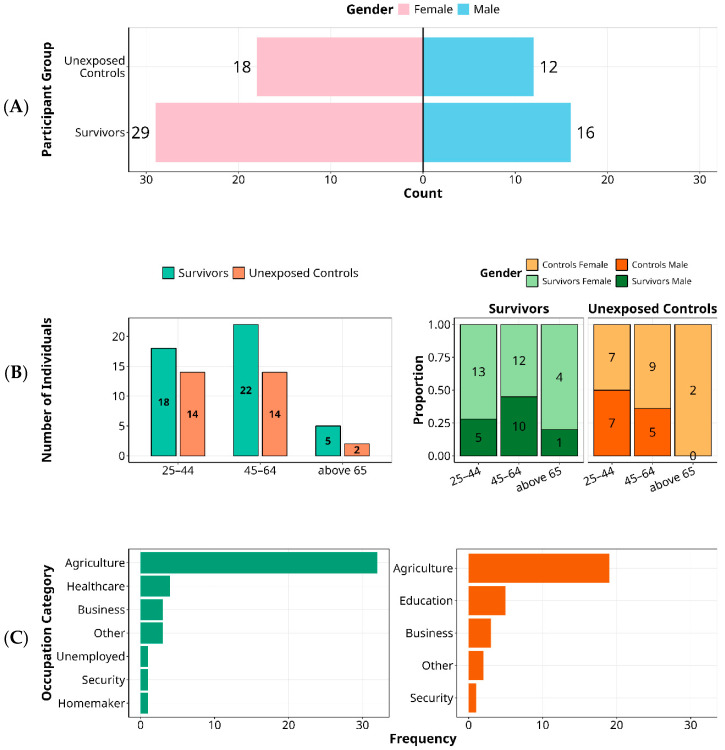
Demographic Characteristics of Ebola Sudan 2000 Survivors and Unexposed Controls by Gender, Age, and Occupation. Key demographic variables among individuals affected by the 2000 Ebola Sudan outbreak, comparing survivors (*n* = 45) to unexposed community controls (*n* = 30) are summarised. Panel (**A**) shows the gender distribution, with a female predominance in both groups. Panel (**B**) presents age distribution at the time of exposure, highlighting a predominance of middle-aged individuals. It further) illustrates age-stratified gender ratios among survivors, showing a decline in the male-to-female ratio with increasing age. Panel (**C**) displays occupational categories, highlighting agriculture as the predominant profession across groups, with healthcare workers represented only among survivors and education professionals more frequent among controls.

**Figure 2 viruses-17-01410-f002:**
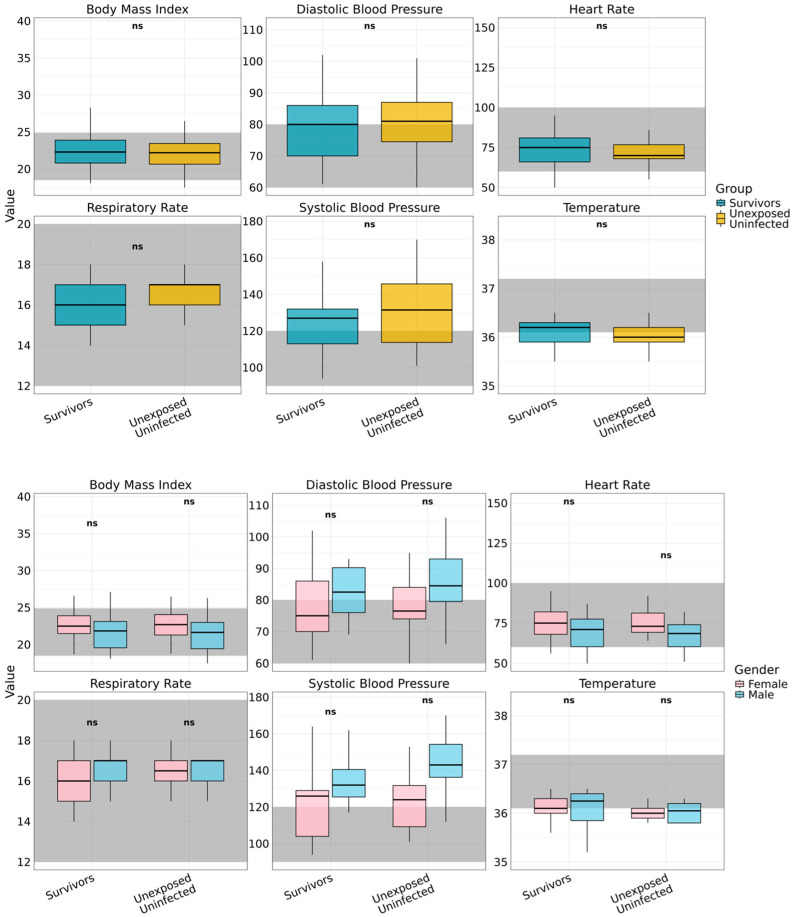
Long-term vital signs assessment by gender and study group. Vital signs assessed among Ebola Sudan virus survivors (*n* = 45) and unexposed controls (*n* = 30), disaggregated by gender are displayed. Evaluated parameters include Body Mass Index (BMI), heart rate, respiratory rate, body temperature, and systolic and diastolic blood pressure. Shaded grey regions indicate established clinical reference ranges for each parameter. Across all measures, no statistically significant gender-based differences were observed within either group (denoted as ns), indicating comparable long-term physiological status between male and female participants.

**Figure 3 viruses-17-01410-f003:**
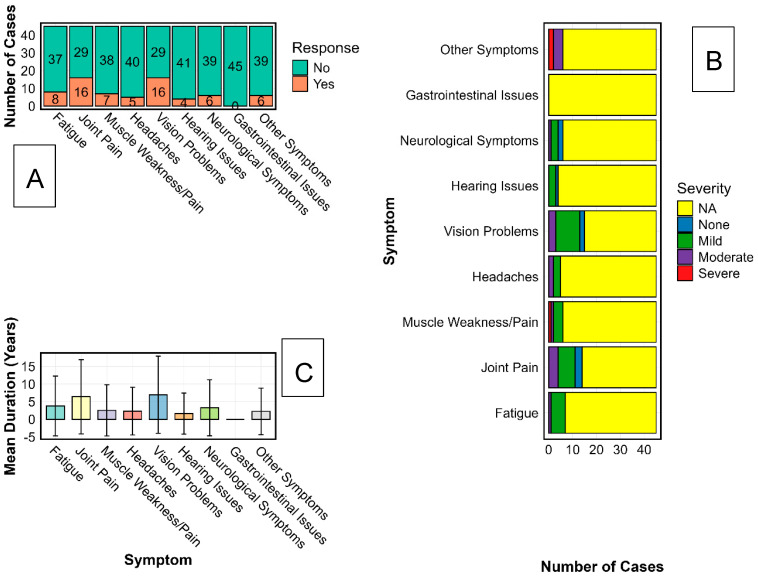
Characterisation of Long-Term Sequelae Among Sudan Virus Disease Survivors. The characterisation of persistent symptoms among Ebola Sudan virus (SUDV) survivors using three analytical approaches is illustrated. Panel (**A**) displays the proportion of survivors reporting chronic symptoms across major categories, including musculoskeletal, sensory, neurological, and systemic complaints. Panel (**B**) presents the distribution of symptom severity, categorised as mild, moderate, or severe, for each symptom type. Panel (**C**) shows the reported duration of symptoms in years, including mean values and standard deviation, across the same symptom categories to indicate variability in long-term health outcomes.

**Figure 4 viruses-17-01410-f004:**
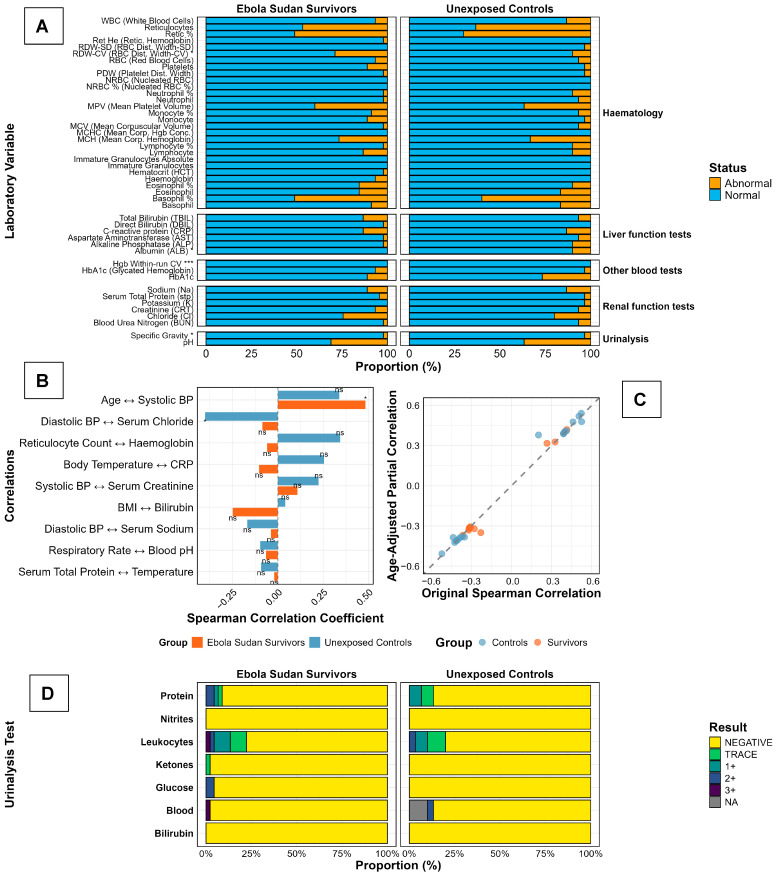
Laboratory Profiles and Vital Sign Correlations in Ebola Sudan Disease Survivors and Unexposed Controls, Two Decades After the 2000 Outbreak. Laboratory findings in Ebola Sudan virus (SUDV) survivors (*n* = 45) and unexposed controls (*n* = 30) are presented. Panel (**A**) displays hematologic, hepatic, and renal parameters, highlighting increased platelet and hemoglobin abnormalities and elevated bilirubin in survivors. Panels (**B**,**C**) depict correlations between clinical and biochemical measures, revealing distinct age-related trends and metabolic associations in survivors, underscoring persistent physiological alterations following SUDV infection. Panel (**D**) shows reticulocyte indices and urinalysis, with notable reticulocyte abnormalities in both groups and mild leukocyturia and hematuria more frequent among survivors. Stacked bars show the distribution of dipstick categories (Negative, Trace, 1+, 2+, 3+; NA = not available) for protein, nitrites, leukocyte esterase, ketones, glucose, blood (haematuria), and bilirubin. TRACE, 1+, 2+, 3+ represent trace, small, moderate, and large amounts, respectively.

**Figure 5 viruses-17-01410-f005:**
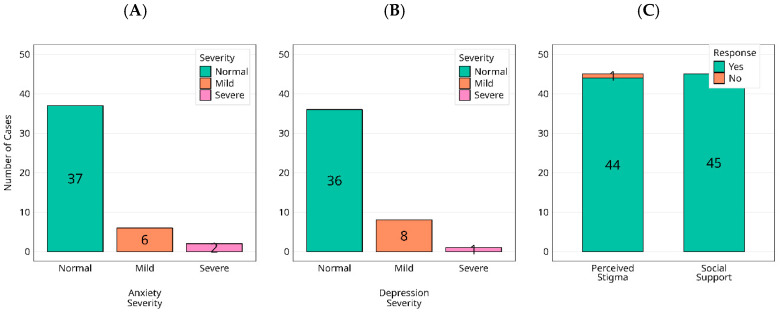
Distribution of current anxiety (**A**) and depression (**B**) categories (normal/mild/moderate/severe), alongside reported experiences of social support and stigmatization (**C**), are presented among individuals exposed to the 2000 Ebola virus outbreak.

## Data Availability

All data has been presented in this manuscript. De-identified participant data are available upon reasonable request directed to the corresponding author jennifer.serwanga@mrcuganda.org. Requests will be considered per the Uganda Virus Research Institute’s data sharing policies.

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
