# Peer review of "Two Decades Later: Long-Term Multisystem Sequelae and Subclinical Organ Dysfunction in Sudan Ebola Virus (SUDV) Survivors of the 2000 Outbreak"

_viruses, 2025, doi:10.3390/v17111410_

Round 1

Reviewer 1 Report

Comments and Suggestions for Authors

As long-term symptoms related to Ebola virus disease (EVD) clearly impact the long-term health of survivors, data is needed on long-term EVD survivor health, and this study evaluates the survivors of the 2000 Sudan virus outbreak - 25 years after infection - and thus represents one of the only studies analyzing survivors that long after infection. The authors have enrolled and assessed long-term sequelae and clinical vital signs and a panel of biochemical assessment of blood and urine in 45 survivors of the 2000 Sudan ebolavirus outbreak in Gulu, Uganda and compared with 30 age and sex-matched controls. They find that survivors have comparable metrics on CBC and metabolic panels as age and sex-matched controls. There were some abnormalities that were identified, related to liver function although the authors could make modifications to the figures to emphasize this point to the readers. The authors find that a subset of SUDV survivors reported sequelae symptoms, although comparative prevalence with controls could not be established as the controls were not assessed/asked about sequelae symptoms but are consistent with other reports of sequelae associated with Ebola virus infection.  The duration of sequelae is potentially informative, but is subject to recall bias, which the authors acknowledge. The subclinical phenotype related to hepatic and platelet biology is based on subtle differences, and I am a little unclear on how the relationship to age supports the claim of hepatic and platelet abnormalities. Overall, it is encouraging that many of the metrics measured by the authors are not different between survivors and controls. I think the manuscript could be improved by 1) clarification on how the sequelae data were collected, and whether it represents symptoms that the survivors are currently experiencing or historic recollection; 2) clearer interpretation of the metabolic data; and 3) revision to figures to help with interpretation.

Main points:

  1. Could the questionnaire that was administered to the survivors be included in the supplemental information?
  2. The duration analysis is interesting and potentially informative for management of survivors of more recent SUDV and EBOV outbreaks regarding how long sequelae symptoms may last. Were any survivors analyzed here still experiencing sequelae symptoms at the time of sampling/examination? Could the duration/cessation of specific symptoms be supported by other clinical findings or historic medical records?
  3. The interpretation of the associations of metabolic/CBC counts with age that are or are not observed in survivors are unclear to me and it is not clearly stated how or why these associations may be indicative of hepatic or platelet abnormalities.
  4. It is unclear whether the mental health screening referred to survivors’ mental health at time of the survey or reflects mental health in the years following the outbreak. If at the time of the survey, inclusion of the controls could be informative to demonstrate that differences between survivors and controls are or are not significant.
  5. Would it be possible to expand upon the section describing the stigma that survivors face while contrasting with the support received after the outbreak? Is it more clinical support that the survivors received? How long did survivors feel supported? Expanding this section in the discussion could further help guide policy and give the readers perspective on the current challenges that survivors face.
  6. The link between viral persistence and specific sequelae has not been clearly established, and so I would caution the authors against making statements regarding specific sequelae being consistent with presence of viral antigen.

Minor comments:

Figure 3.

  1. For panel A, it would be helpful to also include the symptoms directly underneath the graph to allow the reader to better grasp which sequelae are reported by survivors.

Figure 4:

  1. The panels are not labeled as described in the figure legend.
  2. It would be helpful to the reader to indicate in panel A the measures that were significantly different between survivors and controls, perhaps by a star or other notation on the right-hand side of the graph.
  3. In panel B (urinalysis test) – it is unclear what 1+, 2+, 3+ indicate.
  4. The data shown in panel C is discordant with the text that describes the data. In the text, the authors mention that age is associated with albumin in survivors, yet the graph indicates that the association is in unexposed controls.
  5. This figure contains a lot of acronyms that would be helpful to clarify in either the figure itself or in the figure legend. While some are indicated (for example total Bilirubin (TBIL), others (for example, MPV and A1-W3) are not.

Reviewer 2 Report

Comments and Suggestions for Authors

This manuscript presents a unique and valuable long-term follow-up of Sudan Ebola virus (SUDV) survivors from the 2000 Gulu outbreak in Uganda. The study leverages detailed clinical, biochemical, and psychosocial assessments in a well-defined cohort, offering rare insights into persistent health outcomes 25 years after infection. The integration of physical, laboratory, and social measures is commendable and provides a nuanced picture of recovery and residual burden. The findings are important for informing long-term care strategies for filovirus survivors and should be of interest to Viruses readers. However, several issues (mostly minor) should be addressed to improve clarity and interpretability.

First, the authors chose not to collect symptom inventories from the control group in order to avoid recall bias. While this decision is methodologically defensible, it limits the ability to assess the specificity of long-term symptoms. This limitation should be more explicitly discussed in both the Methods and Discussion sections, and it should be clearly stated that conclusions about symptom prevalence are limited to the survivor cohort.

Second, the Discussion emphasizes possible viral persistence and immune-mediated tissue injury in immune-privileged sites. However, no virological testing (e.g., viral RNA or antigen detection) or immunophenotyping was conducted. This gap should be acknowledged more directly, and the pathogenetic hypotheses should be framed as speculative, pending future mechanistic studies.

Third, some of the observed abnormalities, such as albumin decline and reduced mean platelet volume, are known to be age-associated. Although age-matching was performed, the potential confounding effect of aging remains difficult to disentangle from SUDV-related sequelae. It would be helpful to clarify whether interaction or stratified analyses were performed to address this, and if not, the authors should explicitly acknowledge this limitation.

Fourth, the demographic characteristics of the survivor cohort should be more directly contextualized with respect to the broader survivor population from the 2000 Gulu outbreak. This would help assess representativeness and potential survivor bias.

Finally, terms such as “subclinical phenotype” and “latent immunopathological signatures” may be difficult for some readers outside the specialty. These should be briefly clarified or replaced with more accessible language where appropriate.

Author Response

This manuscript presents a unique and valuable long-term follow-up of Sudan Ebola virus (SUDV) survivors from the 2000 Gulu outbreak in Uganda. The study leverages detailed clinical, biochemical, and psychosocial assessments in a well-defined cohort, offering rare insights into persistent health outcomes 25 years after infection. The integration of physical, laboratory, and social measures is commendable and provides a nuanced picture of recovery and residual burden. The findings are important for informing long-term care strategies for filovirus survivors and should be of interest to Viruses readers. However, several issues (mostly minor) should be addressed to improve clarity and interpretability.

We sincerely thank the Reviewer for the thoughtful and constructive evaluation of our manuscript on long-term health outcomes among Sudan Ebola virus (SUDV) survivors from the 2000 Gulu outbreak. We appreciate the recognition of our cohort's uniqueness, the integration of clinical, laboratory, and psychosocial assessments, and the relevance of our findings for survivor care. In response to the Reviewer's comments, we have revised the manuscript to improve clarity and interpretability, as detailed below. We believe these changes strengthen the presentation and ensure the work is maximally useful to Viruses readers.

  1. First, the authors chose not to collect symptom inventories from the control group in order to avoid recall bias. While this decision is methodologically defensible, it limits the ability to assess the specificity of long-term symptoms. This limitation should be more explicitly discussed in both the Methods and Discussion sections, and it should be clearly stated that conclusions about symptom prevalence are limited to the survivor cohort.

We thank the reviewer for this important point. Our design intentionally restricted detailed symptom inventories to SUDV survivors to minimise differential recall bias between survivors (who underwent disease-specific follow‑up) and controls (who lacked an equivalent illness anchor). In response, we now (i) explicitly justify this choice in the Methods (Sections 2.2 and 2.5), (ii) state in Results/Discussion that between‑group estimates of symptom prevalence were not undertaken and that inference on symptom burden is restricted to survivors, and (iii) expand the limitations to emphasise implications for specificity and future study design. We also add a clarifying sentence in the Abstract. Revised text is marked below and incorporated in the tracked‑changes/redline version with line numbers.

Where edited:

Methods §2.2 (L136-L141), §2.5 (L190-L193),

Results §3.3 (L260-L262),

Discussion (L482-L497)

Second, the Discussion emphasizes possible viral persistence and immune-mediated tissue injury in immune-privileged sites. However, no virological testing (e.g., viral RNA or antigen detection) or immunophenotyping was conducted. This gap should be acknowledged more directly, and the pathogenetic hypotheses should be framed as speculative, pending future mechanistic studies.

We agree and have revised the Abstract (Interpretation), Results (§3.3), and Discussion to explicitly acknowledge that we did not perform virological testing or immunophenotyping, and that any pathogenetic interpretations (e.g., antigen persistence or immune-mediated injury) are speculative. We now describe these as hypothesis-generating signals and outline concrete future mechanistic work (targeted virological assays of immune-privileged compartments where ethically feasible, longitudinal imaging/ophthalmic evaluations, and deep immunophenotyping of cryopreserved PBMCs). We also added a strengthened Limitations paragraph reiterating these points.

Where edited:

Abstract (L43-L45)

Results §3.3 (L280-L282),

Discussion (L457-L459) and (L489-493)

Third, some of the observed abnormalities, such as albumin decline and reduced mean platelet volume, are known to be age-associated. Although age-matching was performed, the potential confounding effect of aging remains difficult to disentangle from SUDV-related sequelae. It would be helpful to clarify whether interaction or stratified analyses were performed to address this, and if not, the authors should explicitly acknowledge this limitation.

Thank you. We addressed age in three ways: (i) group-level age-matching at enrolment; (ii) age-adjusted partial Spearman correlations for all vital-sign–laboratory pairs; and (iii) sensitivity summaries showing that age adjustment produced minimal change (all previously significant correlations retained; mean |Δr|≈0.02). We have now made this more explicit in the Methods and Results, and we added a new Figure 4C (raw vs age-adjusted r). We did not perform formal group×age interaction tests or age-stratified comparisons due to limited power and risk of unstable estimates; this is now stated plainly in the Limitations.

Where edited:

Methods §2.5  (L194-L198)

Results §3.4 (L339-L343),

Discussion  (L509-512)

Fourth, the demographic characteristics of the survivor cohort should be more directly contextualized with respect to the broader survivor population from the 2000 Gulu outbreak. This would help assess representativeness and potential survivor bias.

Thank you for this helpful suggestion. We have now contextualised our survivor cohort against the broader 2000–2001 Gulu SUDV outbreak and clarified potential sources of survivor bias.

Where edited:

Methods §2.1 (L114-L122)

Finally, terms such as “subclinical phenotype” and “latent immunopathological signatures” may be difficult for some readers outside the specialty. These should be briefly clarified or replaced with more accessible language where appropriate.

We thank the reviewer for highlighting that terms such as “subclinical phenotype” and “latent immunopathological signatures” could be difficult for nonspecialist readers. We agree and have revised the manuscript to use plainer, explanatory language and to define these concepts at the first relevant mention. We harmonized wording so that references to “subclinical” effects consistently use the same plainlanguage definition.

We believe these changes improve accessibility without sacrificing scientific precision. We appreciate the reviewer’s suggestion.

Summary of changes:

1) Replaced jargon with plain language:

  • “latent immunopathological signatures” → “subtle immune-related changes detectable only on laboratory testing” (Introduction; see line numbers 95; Methods §2.3 L153 ).
  • “subclinical phenotype” → “pattern of mild, symptomfree laboratory changes”

(Results 3.4 and Results 3.4; see line number 354

Discussion; see line numbers 460).

Round 2

Reviewer 1 Report

Comments and Suggestions for Authors

The authors have addressed my questions and I think their edits have greatly improved the manuscript. 

Reviewer 2 Report

Comments and Suggestions for Authors

All my comments have been addressed.